# Pre- and post-migration determinants of self-rated health among Ukrainian refugees in Germany: A cross-sectional comparative analysis with recently arrived refugees from other countries of origin

Louise Biddle[1], Andrea Marchitto[1], Sabine Zinn[1,2]*

**1** Socio-Economic Panel, German Institute for Economic Research (DIW Berlin), Berlin, Germany,
**2** Institute of Social Sciences, Humboldt University Berlin, Berlin, Germany

* szinn@diw.de

## Abstract

6.5 million Ukrainian refugees have been displaced globally since 2022, with one million who registered for temporary protection in Germany under the EU Temporary Protection Directive. Unlike other refugee groups, they were granted immediate access to social security and health care. However, little is known about the differences in health determinants for individuals arriving under the EU Temporary Protection Directive versus those seeking protection on the basis of asylum law, limiting the evidence base for policy intervention. Thus, a comparative analysis is needed. We use a representative survey of Ukrainian refugees in Germany (2023) to analyse the effect of pre- and post-migration factors on self-rated health using multiple logistic regression (n = 5943). We contrast these findings with identical analyses among non-Ukrainian refugees who arrived in Germany mainly during 2015/16 and were interviewed within one year after their arrival (n = 1195). In the Ukrainian sample, post-migration factors are particularly critical for health, with those experiencing discrimination (OR: 1.9, 95%CI: 1.6 – 2.3) and social isolation (OR: 2.7, 95%CI: 2.2 – 3.2) affected by ill health, while those attending a German language course (OR: 0.7, 95%CI: 0.6 – 0.9), with "sufficient" German proficiency (OR: 0.7, 95%CI: 0.6 – 1.0), and frequent contact with Germans (OR: 0.7, 95%CI: 0.5 – 0.8) have better health. Pre-migration factors do not affect self-rated health. Among non-Ukrainian refugees, pre- and post-migration factors are not associated with health, apart from social isolation (OR: 2.2, 95%CI: 1.4 – 3.2). Despite favourable legal entitlements, the health of Ukrainian refugees in Germany is shaped by adverse post-migration circumstances. This finding underscores the importance of expanding diversity-sensitive healthcare approaches, including outreach services and medical interpreters. For non-Ukrainian refugees, restrictive legal conditions pose substantial health risks that become more evident over time.

**Data availability statement:** Data of the Socioeconomic Panel is publicly available for individuals at research institutions subject to eligibility requirements and data sharing agreements. Requests can be made via soepmail@diw.de.

**Funding:** This study was financially supported by the German Science Foundation (DFG) in the scope of the SUARE project (project number 518967487). No additional external funding was received for this study. The funder had no role in study design, data collection and analysis, decision to publish, or preparation of the manuscript.

**Competing interests:** The authors have read the journal's policy and have the following competing interests: LB reports a previous consulting relationship with the Robert Koch-Institute. There are no patents, products in development or marketed products associated with this research to declare. This does not alter our adherence to PLOS policies on sharing data and materials.

## 1. Introduction

The ongoing war of Russia against the Ukraine has caused the largest displacement of refugees within Europe since World War II. While a large proportion of people has been displaced within Ukraine, an estimated 6.5 million Ukrainian refugees have sought protection outside of the country as of July 2024 [1]. The largest intake of Ukrainian refugees has been recorded in Germany (1.2 million refugees) and Poland (1 million refugees) [1].

The refugee movement from the Ukraine is unique in Europe in several regards. Firstly, the national conscription of male Ukrainians of working age has meant that the refugee population is largely comprised of women and children, with a high degree of family separation [2]. This stands in contrast to refugee migration from other countries of origin, which has seen a higher proportion of male refugees entering the Europe. Secondly, the geographical proximity of Ukraine to the main destination countries has allowed for relatively easy migration by train and bus. Whereas journey of other refugee groups in Europe is often highly perilous and costly, the ease of travel from Ukraine means that circular migration journeys and temporary stays are more commonplace among Ukrainians [3], allowing for relative proximity to their homeland, family, and friends.

A further feature which makes the situation of Ukrainians in Europe unique is the EU Temporary Protection directive, which was established in 2001 but was activated for the first time in March 2022 in response to the war in Ukraine. This directive granted a temporary residence permit for Ukrainians in the EU, the validity of which has been extended until March 2025. In Germany, the directive has enabled access to social and health services on par with German citizens, even though access to such services remains restricted for refugees from other countries. It has also granted immediate access to the labour market and language courses [4]. Nevertheless, the integration of Ukrainian refugees in Germany has been accompanied by challenges. Despite high levels of education among refugees from Ukraine, only 18% of refugees were employed six months after arrival [4]. Low levels of German language skills and a low rate of pre-school childcare attendance presented barriers to labour market integration, particularly for Ukrainian refugee women with dependent children [4]. At the same time, those in employment tend to be overqualified for the economic opportunities available [4]. Just under half of the Ukrainian refugees reported needing support with German language acquisition, while a third reported needing help finding employment and accessing healthcare, respectively [4].

Very little is currently known about the health situation of Ukrainians who have fled following the invasion of 2022. Given the high burden of chronic illnesses in Ukraine, particularly of cardiovascular illness and diabetes, it can be expected that manging these illnesses is one of the key challenges in healthcare provision to Ukrainian refugees [5]. Existing studies report an increased burden of mental health issues among Ukrainian refugees [6–9]. However, existing studies of the health of Ukrainian refugees are limited by small sample sizes [7] and non-representative sampling frames [6–8]. To date, only two more systematic studies have been conducted in Czechia: a nationally representative survey showing a very high prevalence of depression and

anxiety, combined with substantial unmet needs for mental health care [9], and a panel survey of highly educated women highlighting the role of chronic disease, depressive symptoms, social isolation, and material deprivation for poor self-rated health [10].

Beyond the burden of illness, very little is currently known about the determinants of health among Ukrainian refugees.

Existing research among other refugee groups in Germany has pointed to the detrimental health effects of post-arrival conditions, including substandard housing [11], lack of employment opportunities [12], insecure legal status, low levels of social integration, and perceived discrimination [13]. Yet Ukrainian refugees differ fundamentally in their demographic composition, migration pathways, and legal entitlements under the Temporary Protection Directive. This raises the critical question of whether established evidence on the health of refugees in Germany can be transferred to this new group. Emerging evidence from convenience samples suggests that among Ukrainian refugees, social support and perceived discrimination may be particularly relevant for mental health [6,7], but such findings have yet to be established in rigorous, nationally representative surveys.

A systematic comparison of Ukrainian refugees with other recently arrived refugees is therefore essential. It allows us to assess whether health determinants identified in other refugee groups, primarily refugees fleeing from the middle East during 2015/2016, remain relevant under the altered legal and social conditions introduced by the Temporary Protection Directive. Not only the legal circumstances, but also the demographic composition of both groups makes a direct extrapolation of existing evidence on this topic challenging. Direct evidence on Ukrainian refugees is therefore needed. To aid our understanding of post-arrival health under varying legal and social conditions, Ukrainian refugees must be compared to other refugee groups within the same timeframe (e.g. within the first year). Such evidence is not only important for designing diversity-sensitive health services for Ukrainians but also for understanding the broader role of reception conditions in shaping refugee health.

This analysis aims to contribute to the evidence base by providing the first analysis of health determinants among Ukrainian refugees in Germany using a large, nationally representative dataset. Exploiting a unique dataset of refugees in Germany, we can compare health determinants of Ukrainian refugees with that of other recently arrived refugees. Our research questions are as follows:

A) Which pre- and post-migration factors are associated with the general health of Ukrainian refugees in Germany?

B) How do associations of pre- and post-migration factors with general health differ to recently arrived refugees from other countries of origin?

Our empirical findings are relevant for understanding the factors that contribute to improved health among Ukrainian refugees. Consequently, our study results are vital to guide future policy efforts.

## 2. Theoretical background

This analysis considers both pre- and post-migration factors to contextualise the health of recently arrived refugees. Both are crucial in responding to the complex health needs of refugee populations.

### 2.1 Pre-migration determinants of refugee health

In the context of forced migration, the events leading to the involuntary departure from one's country of origin represent the most immediately relevant pre-migration factor for health. It is well known that both the direct and indirect experience of war, violence, and conflict negatively affects mental health [13–15]. However, the relationship between pre-migration experiences and self-rated health is less clear. It is conceivable that the impact on mental health will also come to bear on general health measures. Furthermore, it could be expected that experiences of war, conflict, and flight will cause additional physical health problems, including injuries, sexual, and reproductive health issues or long-term effects on cardiovascular health. On the other hand, it is also conceivable that we see an improvement in subjective health in individuals

from particularly badly affected regions due to different frames of reference: comparing themselves to others in the region they fled from, refugees from highly violent areas may count themselves particularly lucky to be in safety and thus evaluate their health more positively. A similar effect could, for example, be seen during the COVID-19 pandemic, when subjective health improved in the general population [16].

## 2.2 Post-migration determinants of refugee health

With respect to the post-migration phase, the existing literature points to factors related to the legal status and reception condition—including the asylum process for most refugee groups, and the Temporary Protection framework for Ukrainian refugees—as well as economic and social integration as crucial determinants of health.

For refugees from countries other than Ukraine, the asylum process in Germany may last from 3 to over 36 months depending on the country of origin and the complexity of the asylum application, during which refugees are obliged to reside in state-provided collective accommodation centres, are subject to residential transfers at short notice, and receive limited social and health benefits [17,18]. Asylum seekers' health care entitlements are restricted to services for acute illnesses and pain, legally mandated prevention measures, and pregnancy-related care [19], and these restrictions have been linked to substantial delays to care, missed treatment, and an increased burden of disease [18]. The economic opportunities of asylum seekers are also limited by both complex bureaucratic processes to obtain work permits and challenges accessing German language learning opportunities [20]. Unemployment is therefore high among non-Ukrainian refugees, with consequences primarily for mental health [13]. With respect to the social networks, forced migration is often accompanied by a stark reduction of close social contacts, including family. After arrival in Germany, collective accommodation centres may represent a crucial social infrastructure, facilitating contacts with other refugees and volunteers/aid workers. However, the forced close proximity of individuals in accommodation centres can also be a source of conflict and violence [21], while the frequent relocation of asylum seekers and remote geographical location of accommodation centres makes the establishment of local social networks difficult. For Ukrainians, the existing diaspora in Germany may provide social contacts and ease the transition into German society. For refugees from other countries of origin, such as Syria or Afghanistan, migration to Germany prior to the refugee movements has been historically lower than from Ukraine [22], meaning that diasporic networks are less likely to be a resource for newcomers.

The post-arrival context is fundamentally shaped by the constraints of the legal asylum system and as such is completely different for Ukrainian refugees and those arriving from other countries. Since Ukrainian refugees have comparable access to the social system as the local population through the provisions of the EU Temporary Protection Directive, it might be anticipated that post-migration factors would have minimal impact on their health. On the other hand, there is now substantial evidence for an "integration paradox" among migrants [23]: as migrants are increasingly integrated into their host societies, they report increasing experiences of discrimination. As new migrants acquire language skills, find employment opportunities, have contacts with members of the majority population, and are thus increasingly "exposed" to the structures, debates, and opinions of members of the host societies they are a) more likely to encounter discriminatory structures and behaviour and b) more likely to frame experiences in terms of discrimination. This effect appears to be mediated by both the phenotypical identifiability of individuals as a migrant (for example by skin colour or hair structure) and the socio-political context of reception, with migrants being more aware of remaining inequities in societies in which marginalisation is smaller (Toqueville's paradox) [23]. Thus, while Ukrainian refugees may benefit from their phenotypical similarity, making them less readily identifiable as an "other", paradoxically their rapid inclusion into German society through the EU Temporary Reception Directive may mean that they experience frequent discrimination. As a result of rapid language acquisition, integration into the labour market, and social contacts with both the existing network of Ukrainian migrants in Germany and with Germans, Ukrainians may be more acutely aware of the current discussions concerning their place in German society, their access to health and social benefits, and the precarity of their legal status. Inequalities with respect to post-arrival conditions may therefore be felt more acutely, with a detrimental impact on health.

This may be compounded by the differential socio-demographic composition of Ukrainian refugees: young, female, highly educated refugees fleeing the war in Ukraine may perceive the post-arrival conditions differently to refugees from other countries of origin with different socio-economic backgrounds. We could therefore form an alternative hypothesis, that in fact the post-migration determinants of health are more immediately important for Ukrainian refugees compared to those from other countries due to their rapid integration into German society.

## 3. Methods

### 3.1 Dataset and sample selection

This study uses a unique data of Ukrainian refugees in Germany, collected as part of the IAB-BiB/FReDA-BAMF-SOEP study in 2022 and 2023 [4]. The study drew a representative sample of 48 000 Ukrainians using a sampling frame of address registries (*Einwohnermeldeamt*), corroborated with the registry of foreigners in Germany (*Ausländerzentral-register*). All sampled individuals were invited to take part in study via postal mail and could complete the survey online. Non-respondents were followed up with paper-and-pencil questionnaires. The longitudinal survey design incorporated two survey waves, the initial wave being conducted from August to October in 2022, the follow-up wave between January to March 2023. The wave 1 response rate was 25% [4], and just over half the participants (60,2%) were followed up in the second wave. Many relevant post-migration factors were only assessed in the second wave. Thus, only data from this wave (n = 5943) serves as the primary data source for this study (Fig 1).

In order to construct a comparison group of refugees from other countries of origin, data from the IAB-BAMF-SOEP study of refugees was used [24]. This study was set up in 2016 with the aim of gathering comprehensive data on the social, economic and health circumstances of refugees entering Germany since 2013. We utilise all available waves of the survey from 2016-2021. To ensure comparability with the Ukrainian sample, only individuals who were surveyed within 12 months after arrival in Germany were included in the sample (n = 1195).

### 3.2 Variables

We use self-rated general health as the primary outcome measure, captured by the question "How would you describe your current state of health" with the answer options: "very good", "good", "satisfactory", "not very good" and "poor". Self-rated general health is a highly valid predictor of morbidity and mortality [25] and has been extensively used in health and demographic surveys in different countries and languages, including refugee populations [26]. For analysis, a binary variable contrasting "not very good"/ "poor" answer options with reports of "very good"/ "good"/ "satisfactory" health was used.

We use the model by Vearey et al. [27] to select pre- and post-migration factors, as well as cross-cutting sociodemographic variables, relevant for the analysis of health in a migration context.

Pre-migration experiences were captured by two key variables: the intensity of conflict in the country of birth captured by the Political Terror Scale (PTS) [28] and the reason for leaving the country. Further, individual-level variables capturing war experiences and trauma were not available in the current dataset. The PTS is an instrument measuring the level of political terror in over 200 countries for each year since 1976. It is rated by 3 institutions: Amnesty International, Human Rights Watch and U.S. State Department, each assigning a value from 1 to 5, with higher values indicating higher levels of political repression, war and conflict. For this analysis, an average value of these three observations per year and country was calculated and matched to each respondent by country of birth and the year of leaving the country. For refugees from Ukraine, two PTS values are available, depending on whether the area is currently under Russian occupation (PTS = 3) or not- (PTS = 0). We therefore matched region of origin in Ukraine and date of leaving Ukraine to data on the occupation of territory in Ukraine (Fig 2) to assign PTS values. Values for non-Ukrainian refugees were collapsed into a binary variable with values PTS<=3 (lower levels of political terror) and PTS > 3 (higher levels of political terror) for comparability between samples. As a further pre-migration variable, we include reasons for leaving the country, comparing those who report war,

 

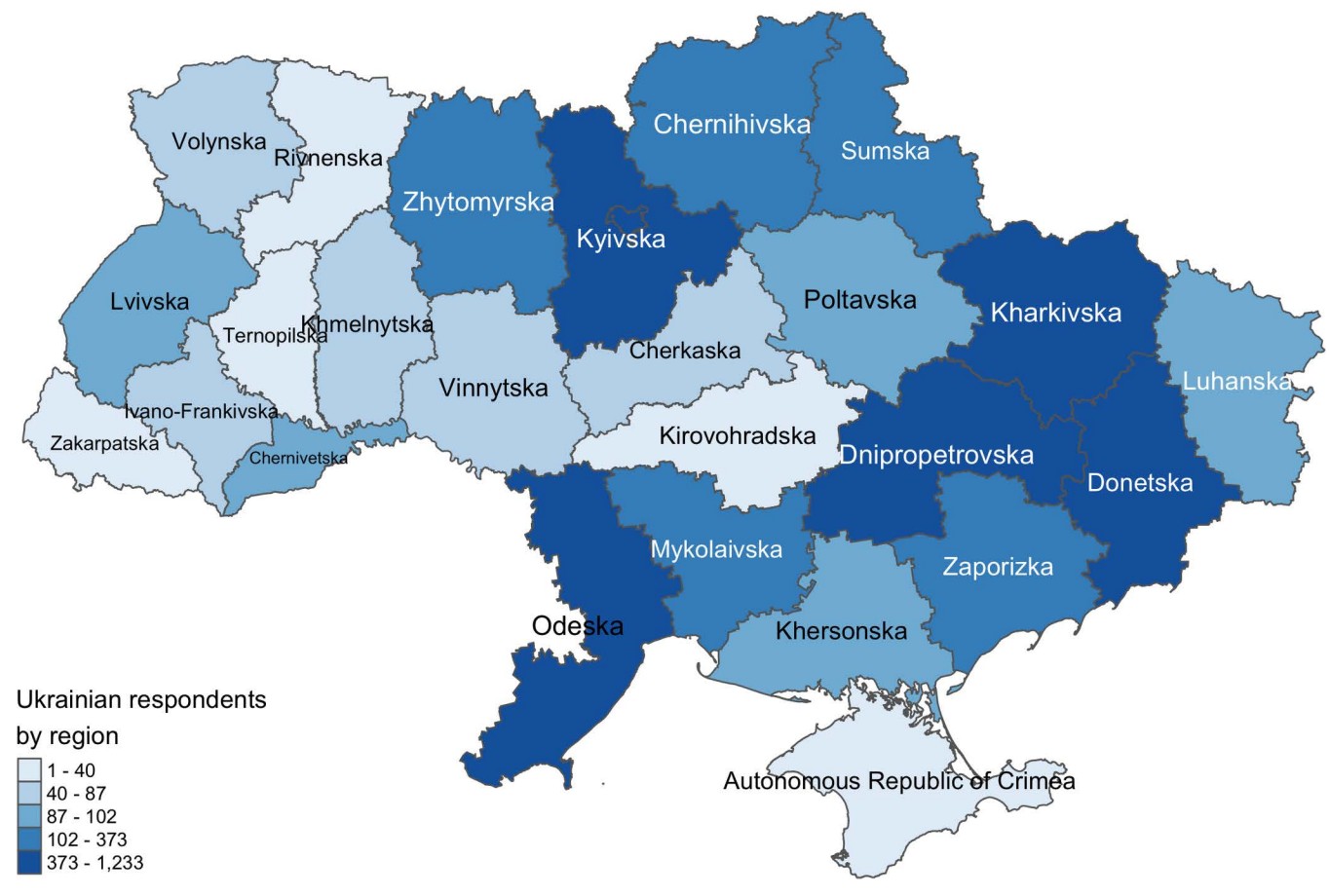

**Fig 1. Number of respondents in the Ukrainian sample by region of origin.** Source: own illustration using data provided by the Humanitarian Data Exchange (OCHA) as map baselayer: https://data.humdata.org/dataset/cod-ab-ukr (Licensed under CC-BY-4.0).

conflict and persecution as reasons for leaving the country with those respondents indicating family- or economic-related reasons only (Table A in S1 File).

In terms of post-migration factors, we include several relevant variables for analysis in the three categories of migration-related, economic and social factors. Migration-related variables include time since arrival (0–6 months and 7–12 months), accommodation type (shared accommodation for refugees, private apartment or house, and other accommodation) and residence status. Because Ukrainian refugees do not enter the asylum process, the residence status variable has been coded differently for the two samples. For the Ukrainian refugees, we contrast those with a temporary protection permit with a) refugees with other visas and b) those without or with unclear residence status. For refugees from other countries of origin, we contrast those with a temporary or permanent resident residence permit with a) refugees awaiting the outcome of their asylum application and b) those with a temporary right to remain ("Duldung") or other status. Because non-Ukrainian refugees are not able to work within the first months after their arrival, we cannot include employment status in our models. To capture economic prospects of individuals in Germany, we include information on attendance of a German or integration course as well as German language proficiency measured by combining information on the self-rated ability to read, speak, and understand German (Table A in S1 File). We further include relevant social factors, including partner's place of residence (no partner, Germany, other

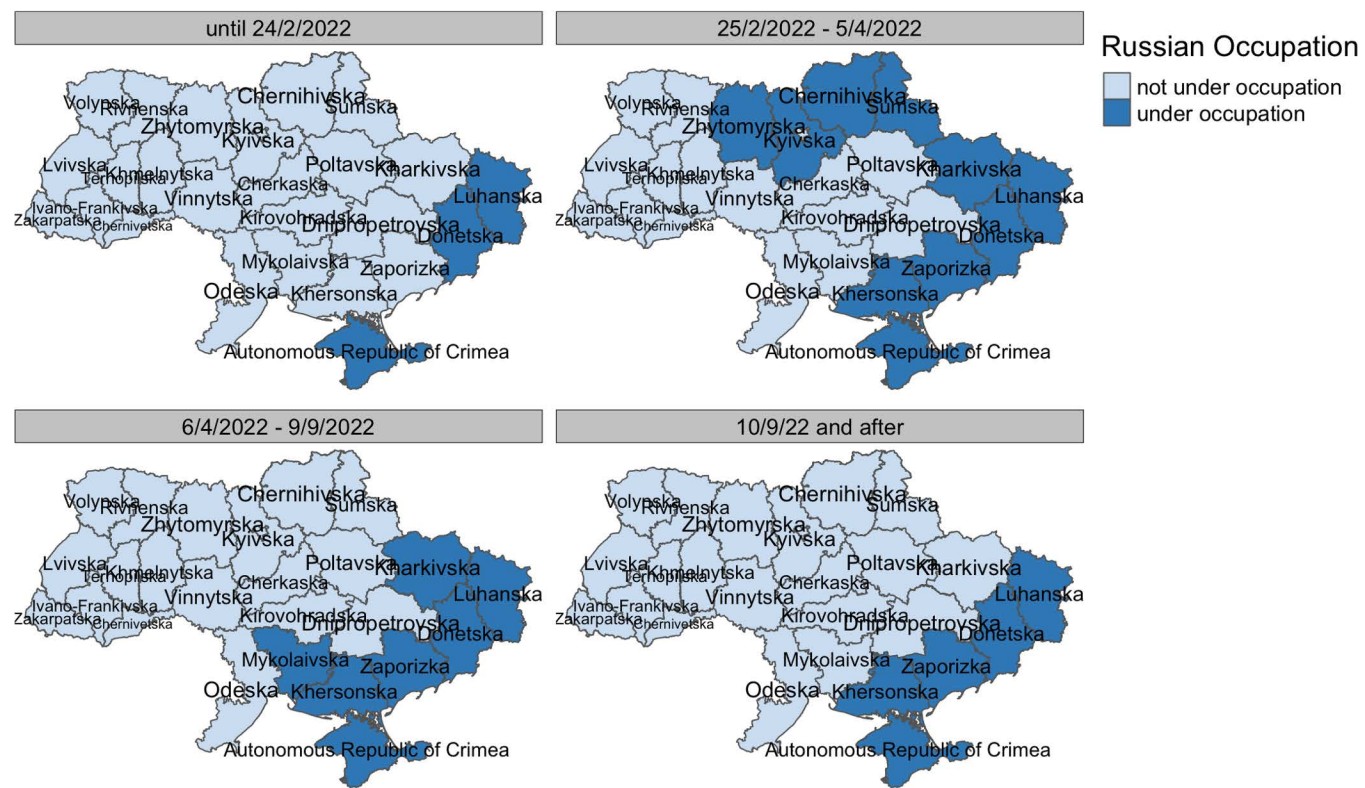

**Fig 2. Occupation of territory in Ukraine at four key timepoints used to match values of the Political Terror Scale to regions of origin of respondents.** Source: own illustration using data provided by the Humanitarian Data Exchange (OCHA) as map baselayer: https://data.humdata.org/dataset/cod-ab-ukr (Licensed under CC-BY-4.0).

country), children's place of residence (no children, all in Germany, one or more living abroad), and perceived discrimination (no experiences of discrimination, some or frequent experiences of discrimination). We also include social isolation, captured as "feeling alone" in the Ukrainian sample and "feeling socially isolated" in the sample of refugees from other countries of origin (Table A in S1 File). Finally, frequency of contact with persons from the same country of origin and with Germans are included (Table A in S1 File).

To account for structural differences in the demographic composition of the two refugee groups, the following socio-demographic variables were considered as covariates in the analysis: sex (male, female), age group (18–30 years, 31–49 years, 50 + years), economic situation before flight (above average, average, below average) and whether the respondent was employed before flight or not. We also include educational status of respondents before flight, categorized as low, medium or high education level (Table A in S1 File). Finally, country of birth is included as an additional covariate in the non-Ukrainian model, distinguishing between Syria, Afghanistan, Iraq, and other countries of birth.

### 3.3 Statistical approach

Bivariate associations between each covariate and self-rated health are calculated by applying Pearson's chi-square tests. We further apply multivariate, logistic regression models to study the association between all health determinants and self-rated health. Model fit is assessed using the p-value of the model F-test and multicollinearity is tested using Variance Inflation Factors (VIF). The proportion of missing values ranges between 0% to 6.3% (Table 1) and we assume that missing values are missing at random (MAR). Missing values are handled by multiple imputation with 30 iterations,

**Table 1. Descriptive overview of all variables used in the main analyses for the Ukrainian and non-Ukrainian refugee samples.**

| | SELF-RATED GENERAL HEALTH | | | | | | | | | | | |
| | Ukrainian refugees | | | | | | Non-Ukrainian refugees | | | | | |
| | good/satisf | | bad | | Total | | good/satisf | | bad | | Total | |
| | No. | % | No. | % | No. | % | No. | % | No. | % | No. | % |
| **Overall distribution** | 5,329 | 89.8% | 603 | 10.2% | 5,932 | 100.0% | 1,037 | 87.0% | 155 | 13.0% | 1,192 | 100.0% |
| | SOCIO-DEMOGRAPHICS | | | | | | | | | | | |
| **Gender** | | | | | | | | | | | | |
| male | 985 | 18.5% | 116 | 19.2% | 1,101 | 18.6% | 620 | 59.8% | 70 | 45.2% | 690 | 57.9% |
| female | 4,339 | 81.4% | 485 | 80.4% | 4,824 | 81.3% | 417 | 40.2% | 85 | 54.8% | 502 | 42.1% |
| . | 5 | 0.1% | 2 | 0.3% | 7 | 0.1% | | | | | | |
| Total | 5,329 | 100.0% | 603 | 100.0% | 5,932 | 100.0% | 1,037 | 100.0% | 155 | 100.0% | 1,192 | 100.0% |
| | Pearson chi2(2) = 2.8281 Pr = 0.243 | | | | | | Pearson chi2(1) = 11.8336 Pr = 0.001 | | | | | |
| **Age group** | | | | | | | | | | | | |
| 18-30 | 1,204 | 22.6% | 135 | 22.4% | 1,339 | 22.6% | 522 | 50.3% | 50 | 32.3% | 572 | 48.0% |
| 31-49 | 2,915 | 54.7% | 250 | 41.5% | 3,165 | 53.4% | 436 | 42.0% | 75 | 48.4% | 511 | 42.9% |
| 50+ | 1,207 | 22.6% | 217 | 36.0% | 1,424 | 24.0% | 79 | 7.6% | 30 | 19.4% | 109 | 9.1% |
| . | 3 | 0.1% | 1 | 0.2% | 4 | 0.1% | | | | | | |
| Total | 5,329 | 100.0% | 603 | 100.0% | 5,932 | 100.0% | 1,037 | 100.0% | 155 | 100.0% | 1,192 | 100.0% |
| | Pearson chi2(3) = 58.9151 Pr = 0.000 | | | | | | Pearson chi2(2) = 30.7637 Pr = 0.000 | | | | | |
| **Economic situation before war** | | | | | | | | | | | | |
| (well) below average | 857 | 16.1% | 153 | 25.4% | 1,010 | 17.0% | 221 | 21.3% | 43 | 27.7% | 264 | 22.1% |
| on average | 2,948 | 55.3% | 318 | 52.7% | 3,266 | 55.1% | 500 | 48.2% | 63 | 40.6% | 563 | 47.2% |
| (well) above average | 1,515 | 28.4% | 131 | 21.7% | 1,646 | 27.7% | 249 | 24.0% | 41 | 26.5% | 290 | 24.3% |
| . | 9 | 0.2% | 1 | 0.2% | 10 | 0.2% | 67 | 6.5% | 8 | 5.2% | 75 | 6.3% |
| Total | 5,329 | 100.0% | 603 | 100.0% | 5,932 | 100.0% | 1,037 | 100.0% | 155 | 100.0% | 1,192 | 100.0% |
| | Pearson chi2(3) = 36.8989 Pr = 0.000 | | | | | | Pearson chi2(3) = 4.8460 Pr = 0.183 | | | | | |
| **Education before flight (level)** | | | | | | | | | | | | |
| low | 717 | 13.5% | 107 | 17.7% | 824 | 13.9% | 619 | 59.7% | 103 | 66.5% | 722 | 60.6% |
| medium | 645 | 12.1% | 106 | 17.6% | 751 | 12.7% | 231 | 22.3% | 22 | 14.2% | 253 | 21.2% |
| high | 3,959 | 74.3% | 389 | 64.5% | 4,348 | 73.3% | 187 | 18.0% | 30 | 19.4% | 217 | 18.2% |
| . | 8 | 0.2% | 1 | 0.2% | 9 | 0.2% | | | | | | |
| Total | 5,329 | 100.0% | 603 | 100.0% | 5,932 | 100.0% | 1,037 | 100.0% | 155 | 100.0% | 1,192 | 100.0% |
| | Pearson chi2(3) = 27.0827 Pr = 0.000 | | | | | | Pearson chi2(2) = 5.2969 Pr = 0.071 | | | | | |
| **Employed before flight** | | | | | | | | | | | | |
| no | 782 | 14.7% | 134 | 22.2% | 916 | 15.4% | 365 | 35.2% | 62 | 40.0% | 427 | 35.8% |
| yes | 4,539 | 85.2% | 468 | 77.6% | 5,007 | 84.4% | 635 | 61.2% | 91 | 58.7% | 726 | 60.8% |
| . | 8 | 0.2% | 1 | 0.2% | 9 | 0.2% | 37 | 3.6% | 2 | 1.3% | 39 | 3.3% |
| Total | 5,329 | 100.0% | 603 | 100.0% | 5,932 | 100.0% | 1,037 | 100.0% | 155 | 100.0% | 1,192 | 100.0% |
| | Pearson chi2(2) = 23.6654 Pr = 0.000 | | | | | | Pearson chi2(2) = 3.1473 Pr = 0.207 | | | | | |
| **Country of birth (only REF)** | | | | | | | | | | | | |
| Afghanistan | / | / | / | / | / | / | 104 | 10.0% | 12 | 7.7% | 116 | 9.7% |
| Iraq | / | / | / | / | / | / | 160 | 15.4% | 32 | 20.6% | 192 | 16.1% |
| Syria | / | / | / | / | / | / | 590 | 56.9% | 80 | 51.6% | 670 | 56.2% |

*(Continued)*

| | SELF-RATED GENERAL HEALTH | | | | | | | | | | | |
| --- | --- | --- | --- | --- | --- | --- | --- | --- | --- | --- | --- | --- |
| | Ukrainian refugees | | | | | | Non-Ukrainian refugees | | | | | |
| | good/satisf | | bad | | Total | | good/satisf | | bad | | Total | |
| | No. | % | No. | % | No. | % | No. | % | No. | % | No. | % |
| Other countries | / | / | / | / | / | / | 183 | 17.6% | 31 | 20.0% | 214 | 18.0% |
| Total | / | / | / | / | / | / | 1,037 | 100.0% | 155 | 100.0% | 1,192 | 100.0% |
| | / | | | | | | Pearson chi2(3) = 4.0876 Pr = 0.252 | | | | | |
| **PRE-MIGRATION FACTORS** | | | | | | | | | | | | |
| **PTS** | | | | | | | | | | | | |
| low (<=3) | 2,971 | 55.8% | 332 | 55.1% | 3,303 | 55.7% | 42 | 4.1% | 12 | 7.7% | 54 | 4.5% |
| high (>3) | 2,358 | 44.2% | 271 | 44.9% | 2,629 | 44.3% | 995 | 95.9% | 143 | 92.3% | 1,138 | 95.5% |
| Total | 5,329 | 100.0% | 603 | 100.0% | 5,932 | 100.0% | 1,037 | 100.0% | 155 | 100.0% | 1,192 | 100.0% |
| | Pearson chi2(1) = 0.1056 Pr = 0.745 | | | | | | Pearson chi2(1) = 4.2494 Pr = 0.039 | | | | | |
| **Reason for leaving the country** | | | | | | | | | | | | |
| Other reasons | 245 | 4.6% | 35 | 5.8% | 280 | 4.7% | 797 | 76.9% | 129 | 83.2% | 926 | 77.7% |
| War, conflict and/or persecution | 5,084 | 95.4% | 568 | 94.2% | 5,652 | 95.3% | 240 | 23.1% | 26 | 16.8% | 266 | 22.3% |
| Total | 5,329 | 100.0% | 603 | 100.0% | 5,932 | 100.0% | 1,037 | 100.0% | 155 | 100.0% | 1,192 | 100.0% |
| | Pearson chi2(1) = 1.7543 Pr = 0.185 | | | | | | Pearson chi2(1) = 3.1558 Pr = 0.076 | | | | | |
| **POST-MIGRATION FACTORS** | | | | | | | | | | | | |
| **Time in Germany since arrival** | | | | | | | | | | | | |
| 0-6 months | 274 | 5.1% | 39 | 6.5% | 313 | 5.3% | 77 | 7.4% | 9 | 5.8% | 86 | 7.2% |
| 6-12 months | 5,055 | 94.9% | 564 | 93.5% | 5,619 | 94.7% | 960 | 92.6% | 146 | 94.2% | 1,106 | 92.8% |
| Total | 5,329 | 100.0% | 603 | 100.0% | 5,932 | 100.0% | 1,037 | 100.0% | 155 | 100.0% | 1,192 | 100.0% |
| | Pearson chi2(1) = 1.9056 Pr = 0.167 | | | | | | Pearson chi2(1) = 0.5279 Pr = 0.468 | | | | | |
| **Type of accommodation** | | | | | | | | | | | | |
| in a shared accommodation for refugees | 375 | 7.0% | 56 | 9.3% | 431 | 7.3% | 636 | 61.3% | 105 | 67.7% | 741 | 62.2% |
| in a private apartment/house | 4,207 | 78.9% | 450 | 74.6% | 4,657 | 78.5% | 391 | 37.7% | 46 | 29.7% | 437 | 36.7% |
| other accommodation (hotel, pension) | 745 | 14.0% | 95 | 15.8% | 840 | 14.2% | 8 | 0.8% | 3 | 1.9% | 11 | 0.9% |
| . | 2 | 0.0% | 2 | 0.3% | 4 | 0.1% | 2 | 0.2% | 1 | 0.6% | 3 | 0.2% |
| Total | 5,329 | 100.0% | 603 | 100.0% | 5,932 | 100.0% | 1,037 | 100.0% | 155 | 100.0% | 1,192 | 100.0% |
| | Pearson chi2(3) = 13.2162 Pr = 0.004 | | | | | | Pearson chi2(3) = 6.3378 Pr = 0.096 | | | | | |
| **Residence status (UKR)** | | | | | | | | | | | | |
| no/unclear | 124 | 2.3% | 14 | 2.3% | 138 | 2.3% | / | / | / | / | / | / |
| other Schengen-Area visum/other resid. permit | 186 | 3.5% | 17 | 2.8% | 203 | 3.4% | / | / | / | / | / | / |
| Fiktionsbescheinigung/temp. protection | 4,861 | 91.2% | 559 | 92.7% | 5,420 | 91.4% | / | / | / | / | / | / |
| . | 158 | 3.0% | 13 | 2.2% | 171 | 2.9% | / | / | / | / | / | / |
| Total | 5,329 | 100.0% | 603 | 100.0% | 5,932 | 100.0% | / | / | / | / | / | / |
| | Pearson chi2(3) = 2.0737 Pr = 0.557 | | | | | | | | | | | |
| **Residence/asylum status (REF)** | | | | | | | | | | | | |
| no/awaiting outcome | / | / | / | / | / | / | 380 | 36.6% | 59 | 38.1% | 439 | 36.8% |
| other residence permit/'Duldung' | / | / | / | / | / | / | 145 | 14.0% | 27 | 17.4% | 172 | 14.4% |

*(Continued)*

| | SELF-RATED GENERAL HEALTH | | | | | | | | | | | |
| | Ukrainian refugees | | | | | | Non-Ukrainian refugees | | | | | |
| | good/satisf | | bad | | Total | | good/satisf | | bad | | Total | |
| | No. | % | No. | % | No. | % | No. | % | No. | % | No. | % |
| residence permit (temporary/permanent) | / | / | / | / | / | / | 451 | 43.5% | 66 | 42.6% | 517 | 43.4% |
| . | / | / | / | / | / | / | 61 | 5.9% | 3 | 1.9% | 64 | 5.4% |
| Total | / | / | / | / | / | / | 1,037 | 100.0% | 155 | 100.0% | 1,192 | 100.0% |
| | | | | | | | Pearson chi2(3) = 5.1157 Pr = 0.164 | | | | | |
| **Partner's living place** | | | | | | | | | | | | |
| no partner | 1,965 | 36.9% | 258 | 42.8% | 2,222 | 37.5% | 330 | 31.8% | 42 | 27.1% | 372 | 31.2% |
| Germany | 1,920 | 36.0% | 208 | 34.5% | 2,128 | 35.9% | 577 | 55.6% | 84 | 54.2% | 662 | 55.5% |
| Ukraine-country of origin/other land | 1,437 | 27.0% | 137 | 22.7% | 1,574 | 26.5% | 118 | 11.4% | 27 | 17.4% | 145 | 12.2% |
| . | 7 | 0.1% | 0 | 0.0% | 7 | 0.1% | 12 | 1.2% | 2 | 1.3% | 14 | 1.1% |
| Total | 5,329 | 100.0% | 603 | 100.0% | 5,932 | 100.0% | 1,037 | 100.0% | 155 | 100.0% | 1,192 | 100.0% |
| | Pearson chi2(3) = 9.8813 Pr = 0.020 | | | | | | Pearson chi2(3) = 5.0808 Pr = 0.166 | | | | | |
| **Children living place (UKR)** | | | | | | | | | | | | |
| no child | 1,579 | 29.6% | 177 | 29.4% | 1,756 | 29.6% | / | / | / | / | / | / |
| one/all in Germany | 2,931 | 55.0% | 297 | 49.3% | 3,228 | 54.4% | / | / | / | / | / | / |
| one/all abroad/Ukraine | 797 | 15.0% | 127 | 21.1% | 924 | 15.6% | / | / | / | / | / | / |
| . | 22 | 0.4% | 2 | 0.3% | 24 | 0.4% | / | / | / | / | / | / |
| Total | 5,329 | 100.0% | 603 | 100.0% | 5,932 | 100.0% | / | / | / | / | / | / |
| | Pearson chi2(3) = 16.3540 Pr = 0.001 | | | | | | | | | | | |
| **Children living place (REF)** | | | | | | | | | | | | |
| no child | / | / | / | / | / | / | 448 | 43.2% | 53 | 34.2% | 501 | 42.0% |
| all in Germany | / | / | / | / | / | / | 458 | 44.2% | 69 | 44.5% | 527 | 44.2% |
| one/all abroad/died | / | / | / | / | / | / | 104 | 10.0% | 32 | 20.6% | 136 | 11.4% |
| . | / | / | / | / | / | / | 27 | 2.6% | 1 | 0.6% | 28 | 2.3% |
| Total | / | / | / | / | / | / | 1,037 | 100.0% | 155 | 100.0% | 1,192 | 100.0% |
| | | | | | | | Pearson chi2(3) = 18.1292 Pr = 0.000 | | | | | |
| **Perceived discrimination** | | | | | | | | | | | | |
| never | 3,236 | 60.7% | 275 | 45.6% | 3,511 | 59.2% | 665 | 64.1% | 98 | 63.2% | 763 | 64.0% |
| seldomly/often | 2,088 | 39.2% | 326 | 54.1% | 2,414 | 40.7% | 331 | 31.9% | 50 | 32.3% | 381 | 32.0% |
| . | 5 | 0.1% | 2 | 0.3% | 7 | 0.1% | 41 | 4.0% | 7 | 4.5% | 48 | 4.0% |
| Total | 5,329 | 100.0% | 603 | 100.0% | 5,932 | 100.0% | 1,037 | 100.0% | 155 | 100.0% | 1,192 | 100.0% |
| | Pearson chi2(2) = 52.9960 Pr = 0.000 | | | | | | Pearson chi2(2) = 0.1279 Pr = 0.938 | | | | | |
| **Attended German language/integration course** | | | | | | | | | | | | |
| no | 1,239 | 23.3% | 211 | 35.0% | 1,450 | 24.4% | 471 | 45.4% | 86 | 55.5% | 557 | 46.7% |
| yes | 3,932 | 73.8% | 379 | 62.9% | 4,311 | 72.7% | 566 | 54.6% | 69 | 44.5% | 635 | 53.3% |
| . | 158 | 3.0% | 13 | 2.2% | 171 | 2.9% | | | | | | |
| Total | 5,329 | 100.0% | 603 | 100.0% | 5,932 | 100.0% | 1,037 | 100.0% | 155 | 100.0% | 1,192 | 100.0% |
| | Pearson chi2(2) = 40.6915 Pr = 0.000 | | | | | | Pearson chi2(1) = 5.4870 Pr = 0.019 | | | | | |

*(Continued)*

| | SELF-RATED GENERAL HEALTH | | | | | | | | | | | |
| --- | --- | --- | --- | --- | --- | --- | --- | --- | --- | --- | --- | --- |
| | Ukrainian refugees | | | | | | Non-Ukrainian refugees | | | | | |
| | good/satisf | | bad | | Total | | good/satisf | | bad | | Total | |
| | No. | % | No. | % | No. | % | No. | % | No. | % | No. | % |
| **German language profieciency** | | | | | | | | | | | | |
| none/poor | 3,839 | 72.0% | 507 | 84.1% | 4,346 | 73.3% | 707 | 68.2% | 121 | 78.1% | 828 | 69.5% |
| sufficient | 999 | 18.7% | 71 | 11.8% | 1,070 | 18.0% | 261 | 25.2% | 28 | 18.1% | 289 | 24.2% |
| good/excellent | 333 | 6.2% | 12 | 2.0% | 345 | 5.8% | 69 | 6.7% | 6 | 3.9% | 75 | 6.3% |
| . | 158 | 3.0% | 13 | 2.2% | 171 | 2.9% | | | | | | |
| Total | 5,329 | 100.0% | 603 | 100.0% | 5,932 | 100.0% | 1,037 | 100.0% | 155 | 100.0% | 1,192 | 100.0% |
| | Pearson chi2(3) = 43.4393 Pr = 0.000 | | | | | | Pearson chi2(2) = 6.3644 Pr = 0.041 | | | | | |
| **Social isolation (statement of 'feeling alone') (UKR)** | | | | | | | | | | | | |
| does not apply (at all)/neutral position | 3,973 | 74.6% | 303 | 50.2% | 4,276 | 72.1% | / | / | / | / | / | / |
| applies (fully) | 1,349 | 25.3% | 297 | 49.3% | 1,646 | 27.7% | / | / | / | / | / | / |
| . | 7 | 0.1% | 3 | 0.5% | 10 | 0.2% | / | / | / | / | / | / |
| Total | 5,329 | 100.0% | 603 | 100.0% | 5,932 | 100.0% | / | / | / | / | / | / |
| | Pearson chi2(2) = 160.5855 Pr = 0.000 | | | | | | | | | | | |
| **Feeling socially isolated (REF)** | | | | | | | | | | | | |
| never/seldomly/sometimes | / | / | / | / | / | / | 773 | 74.5% | 92 | 59.4% | 865 | 72.6% |
| often/very often | / | / | / | / | / | / | 207 | 20.0% | 56 | 36.1% | 263 | 22.1% |
| . | / | / | / | / | / | / | 57 | 5.5% | 7 | 4.5% | 64 | 5.4% |
| Total | / | / | / | / | / | / | 1,037 | 100.0% | 155 | 100.0% | 1,192 | 100.0% |
| | | | | | | | Pearson chi2(2) = 20.5026 Pr = 0.000 | | | | | |
| **Contact with Ukrainians/ persons from same country of origin (non-relative)** | | | | | | | | | | | | |
| no/seldomly | 998 | 18.7% | 176 | 29.2% | 1,174 | 19.8% | 321 | 31.0% | 53 | 34.2% | 374 | 31.4% |
| often | 1,312 | 24.6% | 131 | 21.7% | 1,443 | 24.3% | 191 | 18.4% | 29 | 18.7% | 220 | 18.5% |
| very often | 2,861 | 53.7% | 283 | 46.9% | 3,144 | 53.0% | 522 | 50.3% | 71 | 45.8% | 593 | 49.7% |
| . | 158 | 3.0% | 13 | 2.2% | 171 | 2.9% | 3 | 0.3% | 2 | 1.3% | 5 | 0.4% |
| Total | 5,329 | 100.0% | 603 | 100.0% | 5,932 | 100.0% | 1,037 | 100.0% | 155 | 100.0% | 1,192 | 100.0% |
| | Pearson chi2(3) = 37.7064 Pr = 0.000 | | | | | | Pearson chi2(3) = 4.2348 Pr = 0.237 | | | | | |
| **Contact with Germans** | | | | | | | | | | | | |
| no/seldomly | 1,956 | 36.7% | 308 | 51.1% | 2,264 | 38.2% | 468 | 45.1% | 79 | 51.0% | 547 | 45.9% |
| often | 1,205 | 22.6% | 131 | 21.7% | 1,336 | 22.5% | 193 | 18.6% | 30 | 19.4% | 223 | 18.7% |
| very often | 2,009 | 37.7% | 151 | 25.0% | 2,160 | 36.4% | 374 | 36.1% | 45 | 29.0% | 419 | 35.2% |
| . | 159 | 3.0% | 13 | 2.2% | 172 | 2.9% | 2 | 0.2% | 1 | 0.6% | 3 | 0.3% |
| Total | 5,329 | 100.0% | 603 | 100.0% | 5,932 | 100.0% | 1,037 | 100.0% | 155 | 100.0% | 1,192 | 100.0% |
| | Pearson chi2(3) = 54.6273 Pr = 0.000 | | | | | | Pearson chi2(3) = 4.0349 Pr = 0.258 | | | | | |

Abbreviations: REF- non-Ukrainian refugee sample; PTS-Political Terror Scale; UKR- Ukrainian sample.

with coefficients combined using Rubin's rule. The impact of multiple imputation is assessed using the Fraction of Missing information (FMI). Statistical significance is assessed at the p < 0.05 level.

To address the uncertainty in measuring conflict intensity at the country of flight and self-rated health, we conduct sensitivity analyses. Specifically, instead of using the PTS instrument, we employ data from the Armed Conflict Location and Event Data (ACLED) [29] database which offers detailed information directly related to the war in Ukraine. We calculate the number of fatalities in each region of Ukraine from the war's onset to the survey respondent's flight date (SA1) as a pre-migration factor (Table B and Figure A in S1 File). Similarly, we use ACLED data to gauge war intensity at the time of interview as an additional post-migration factor (SA2), operationalized as total fatalities in the 7 days before the interview. To assess the validity of our outcome variable, we use "worries about health" as an alternative outcome measure in the Ukrainian sample, comparing those with significant worries to those with few or no worries (SA3). Additionally, we test the validity of our self-rated health outcome variable by using an alternative categorization, comparing poor/not very good/ satisfactory responses to good/very good health (SA4). As a further sensitivity check, we conduct all regression analyses on the main models without imputation (SA5). Moreover, we distinguish different Ukrainian migration cohorts to Germany, adding information on whether migration took place before or after June 2022 (SA6). Considering that most of the non-Ukrainian refugee sample is from Syria, we restrict our sample to Syrian nationals for a more homogeneous group (SA7). We further run a model stratified by educational status to check for any differential effects in both refugee samples (SA8). We also test for potential interactions between the key post-migration determinants of social isolation, discrimination and language proficiency with both gender and age group to further interrogate differential effects in both samples (SA9).

All Analyses were carried out using STATA v18. The corresponding source code is freely available at https://github.com/ bieneSchwarze/RefugeeHealthGermanyUkrainiansVsOthers.

### 3.4 Ethics statement

The SUARE study was approved by the ethics commission of the Federal Institute for Employment Research (IAB; Project 4219). Consent was given by virtue of participation in the study. No further consent for analysis was required due to the anonymisation of data prior to analysis.

## 4. Results

### 4.1. Descriptives

The sample of Ukrainian refugees is comprised predominantly of women (81.3%) and just over half the sample is aged 31–49 years (53.4%). Most Ukrainian refugees have a high level of education (73.3%) and were employed before flight (84.4%). At survey time, most Ukrainians lived in private accommodation (78.5%), with only 7.3% housed in shared accommodation for refugees. 26.5% of the sample reported being separated from their partner and 15.6% reported being separated from a child. 72.7% of Ukrainians reported attending a language or an integration course in Germany, with 73.3% reporting poor German language skills (Table 1).

Refugees from other countries of birth came predominantly from Syria (56.2%), Iraq (16.1%), and Afghanistan (9.7%). 57.9% of this sample was male and almost half are aged 18–30 years. Other notable differences to the Ukrainian sample include a higher proportion with low education (60.6%), a lower proportion employed prior to migration (60.8%), and a higher proportion living in shared accommodation for refugees (62.2%). While just over half of refugees reported attending a language or an integration course (53.3%), 69.5% reported poor German language proficiency (Table 1).

10.2% of the Ukrainians report a bad or very bad health status within the first year of arrival in Germany, while this figure is 13.0% for refugees from other countries of origin during the same period. In the Ukrainian sample, bivariate tests of association show a statistically significant relationship between bad health and several factors: age group, pre-war economic situation, level of education, and employment status before flight. Significant relationships also emerge post-migration conditions, including type of accommodation, partner and child's living place, perceived discrimination,

attendance of language course, German language proficiency, social isolation and measures of social interaction (Table 1). For refugees from other countries of origin, self-rated health is likewise significantly related to sociodemographic characteristics such as gender, age group). In this group, the level of political terror (PTS) in the country of origin shows also a statistically significant association with health. Among post-migration factors, significant associations are found with the child's living place, attendance of language course, German language proficiency, and the feeling of social isolation.

### 4.2 Regression analyses

For the Ukrainian sample, the associations found in the bivariate statistical testing are confirmed in the logistic regression model: respondents over the age of 50 have an increased odds of poor health as compared to 18–30 year olds (OR: 2.1, 95%CI: 1.5−2.9) while those with an above average economic situation before flight (OR: 0.6, 95%CI: 0.5−0.8), and employment before flight (OR: 0.7, 95%CI: 0.6−0.9) have lower odds of poor health as compared to their reference groups (Fig 3, Table C in S1 File). The two pre-migration factors of our analysis (i.e. intensity of conflict and war and/or persecution as a reason for leaving the country) do not show a statistically significant association with self-rated health in the fully adjusted model for the Ukrainian sample (Fig 3). Among the post-migration factors, several variables emerge as being relevant. While those experiencing perceived discrimination (OR: 1.9, 95%CI: 1.6−2.3) and feelings of social isolation (OR: 2.7, 95%CI: 2.2−3.2) have higher odds of poor health, respondents attending a German language course (OR: 0.7, 95%CI: 0.6−0.9), with "sufficient" German language proficiency (OR: 0.7, 95%CI: 0.5−1.0), frequent contact with Ukrainians (OR: 0.8, 95%CI: 0.6−1.0) and Germans (OR: 0.7, 95%CI: 0.5−0.8) have lower odds of poor health.

Risk of multicollinearity is low with VIFs of included variables ranging from 1.01 to 1.72 (Table D in S1 File). Including variable groups (pre- and post-migration factors) one at a time did not alter the results for the Ukrainian sample (Table C in S1 File). The FMI was 3.5% in the final model of the Ukrainian sample, indicating the variance attributable to missing data was small. The F-test indicated that the variables included in the full models were well suited to for the analysis (p<0.0001).

In the non-Ukrainian sample, female respondents (OR: 1.9, 95%CI: 1.2−2.9) and older respondents (OR 50+years: 4.1, 95%CI: 2.1−7.9) have higher odds of poor health, while the evidence for other socio-demographic factors is less consistent than among Ukrainians (Fig 4 and Table E in S1 File). Similarly to the Ukrainian sample, pre-migration factors show no statistically significant associations with health in the fully adjusted models. Of the post-migration factors, frequent feelings of social isolation show a statistically significant relationship with poor self-rated health in the non-Ukrainian sample (OR: 2.2, 95%CI: 1.4−3.2).

In the non-Ukrainian sample, risk of multicollinearity is low with VIFs of included variables ranging from 1.04 to 1.49 (Table D in S1 File). Including variable groups (pre- and post-migration factors) one at a time did not alter the results in the model for non-Ukrainian refugees (Table E in S1 File). The Fraction of Missing Information (FMI) was 6.1% in the final model of the non-Ukrainian sample. The F-test indicated a good model fit at p<0.0001.

Sensitivity analyses broadly confirm the observed results. We continue to see no significant relationship between war intensity and health when using the number of fatalities prior to departure instead of PTS in the Ukrainian sample (SA1, Table F in S1 File). We also do not see a relationship between the number of fatalities in the 7 days prior to the interviews (SA2, Table F in S1 File) or the migration cohort (SA6, Table G in S1 File) and health. When changing the outcome to the indicator "worries about health" in the Ukrainian sample (SA3, Table F in S1 File), the relationships with the pre- and post-migration factors remain consistent. When changing the categorisation of the self-rated health outcome (SA4, Table F in S1 File) we do see a change in the Ukrainian sample with respect to pre-migration factors: those who fled from an area with a PTS value above 3 have increased odds of poor health (OR: 1.3, 95%CI: 1.1−1.4). The models with non-imputed data (SA5, Tables F and H in S1 File) showed similar results to the regression analyses based on imputed data, but with higher accuracy. Stratification of the results by education do not change key relationships between post-migration factors and health in the Ukrainian samples (Tables I and J in S1 File). Similarly, including interaction terms does not

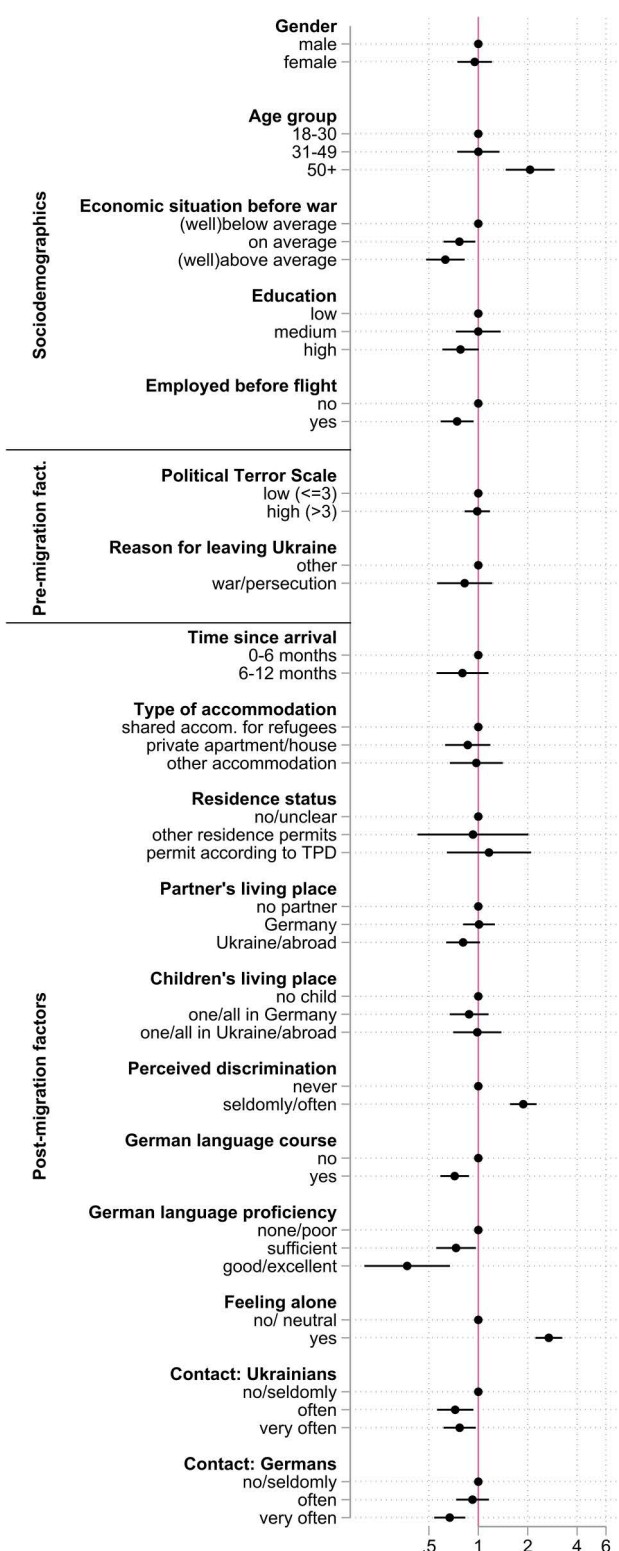

**Fig 3. Results of the fully adjusted logistic regression model for the Ukrainian sample.** Outcome: self-rated general health (poor/not very good vs. very good/good/satisfactory). Note: Circles denote Odds ratios; capped whiskers indicate 95% confidence intervals Abbreviations: accom. – accommodation, TPD – Temporary Protection Directive.

**Global Public Health**

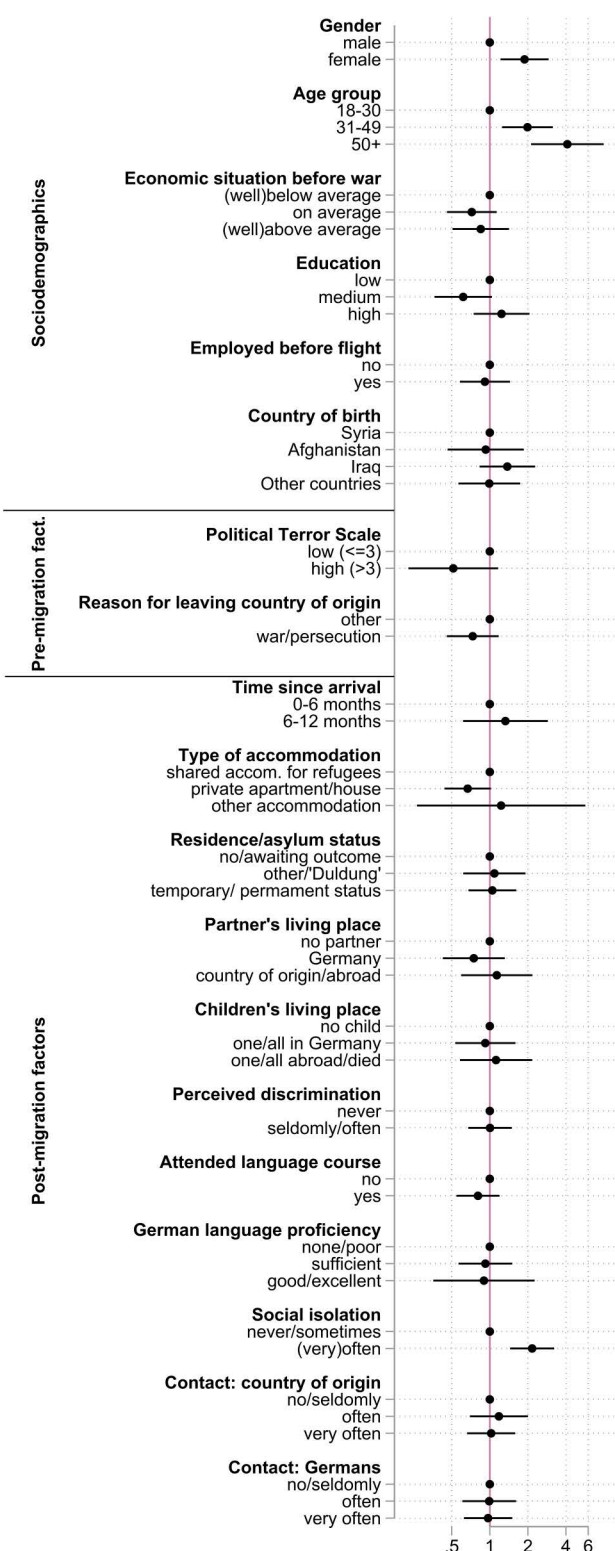

**Fig 4. Results of the fully adjusted logistic regression model for the non-Ukrainian sample.** Outcome: self-rated general health (poor/not very good vs. very good/good/satisfactory). Note: Circles denote Odds ratios; capped whiskers indicate 95% confidence intervals Abbreviations: accom. – accommodation.

change these relationships (Tables K and L in S1 File). Overall, all sensitivity analyses conducted broadly confirm results of the main models (Tables C and E in S1 File).

## 5. Discussion

This study is the first nationally representative study of the pre- and post-migration determinants of health among Ukrainian refugees. By situating these findings in direct comparison with non-Ukrainian refugees, we highlight how different legal and social conditions of arrival shape health trajectories. This comparative perspective is essential, as Ukrainians entered Germany with immediate legal entitlements under the EU Temporary Protection Directive, whereas non-Ukrainian refugees faced prolonged asylum processes and restricted access to social systems. These structural differences may alter how quickly post-migration circumstances affect health.

We find that pre-migration war experiences are not related to self-rated health in both Ukrainian and non-Ukrainian refugee samples within the first year of settling in Germany. This suggests that, at least in the early post-migration phase, conditions in the host country outweigh prior war experiences in shaping subjective health. Post-migration circumstances, including experiences of discrimination, social isolation and language skills, are very much intertwined with the self-rated health of Ukrainian refugees, but this is not case for non-Ukrainian refugees within the first year of arrival.

Our study finds strong evidence of the relationship between post-migration factors, especially subjective experiences of discrimination and loneliness, and the subjective health of Ukrainian refugees. Interaction analysis reveals that, among Ukrainian refugees, perceived discrimination has a significantly stronger negative effect on self-rated health in women than in men. Additionally, the negative impact of social isolation on health diminishes notably with age. These results indicate that despite the extensive entitlements granted by the EU Temporary Protection Directive, Ukrainians face challenges of discrimination and loneliness that directly translate into poorer health outcomes. This is in line with other studies exploring post-migration circumstances for Ukrainian refugees in other European countries: while loneliness and discrimination was found to affect mental health in a two multi-country studies [6,7], poor social contacts have been linked to poor self-rated health among Ukrainian women in Czechia [10]. In the non-Ukrainian sample, on the other hand, we do not observe a relationship between post-migration determinants and subjective health, except for feelings of social isolation, within the first year after arrival in Germany. Given the cross-sectional nature of our analysis, and the manifold demographic and social differences between both groups, we cannot provide causal evidence for the reasons behind these differences. However, the contrast suggests that immediate access to participation rights under the Temporary Protection Directive may provide earlier exposure to both opportunities (e.g., language learning, social contact) and vulnerabilities (e.g., discrimination, isolation), while for non-Ukrainian refugees, the health effects of post-migration conditions emerge only after longer residence. This interpretation aligns with earlier studies of refugees in Germany, some of which have used the same underlying dataset, which find evidence for a strong correlation between a much wider range of post-migration determinants and health when including refugees that have been in Germany for a longer time period [12,13].

The evidence therefore supports the integration paradox for the subjective health of refugees in Germany: while post-migration determinants become increasingly important for non-Ukrainian refugees in the course of their integration into German society, these factors are more immediately important for Ukrainian refugees, who have been granted the possibility of "accelerated" participation under the EU Temporary Protection Directive. While the access granted to social systems has been instrumental for facilitating the integration of Ukrainian refugees in Germany, the directive alone is not sufficient in safeguarding the health of newly arrived Ukrainians. Thanks to the directive, legal health care entitlements are on par with the German population, but equitable access must be ensured by adopting diversity-oriented models of care [30]. Crucially, this includes participative health communication to aid Ukrainians in navigating the highly complex German health system and its bureaucratic access hurdles. It also requires adequate translation and interpretation services, especially to facilitate utilisation of mental health services. Outreach services, such as community health nursing models [31], should also be considered to raise awareness of existing services and provide low-threshold pathways into appropriate care structures.

With respect to the pre-migration determinants of health, the picture presented by the current analysis is less clear: while we find no association between war and/or persecution as reasons for leaving the country with subjective health in our models, a statistically significant relationship with such reports exists when adopting a more inclusive definition of ill health in the Ukrainian samples. However, the observed relationships go in different directions; respondents who report war experiences as a reason for leaving the country have lower odds of poor health while those whose region of origin was badly affected have higher odds of poor health. Existing research on the health of Ukrainian refugees in Norway suggests that the reason more recent arrivals may be in poorer health may be due to prolonged exposure to the war [32], but the causality of this relationship has yet to be established. Further research with more specific pre-migration variables is needed to clarify diverging relationships and the lack of a relationship for non-Ukrainian refugees.

Strengths of the current study include the rigorous, representative approach to data collection, the wide variety of post-migration factors included, and the sample size obtained. Furthermore, the similarity of data collection approach and survey items between the Ukrainian and non-Ukrainian samples means that we can build analytical models which allow for comparisons between the two samples. A further strength is the use of linkage to external data sources on political terror and war fatalities at a regional level in the Ukrainian sample. Finally, while most research reports health determinants several years after arrival, our analysis specifically looks at the first year post-arrival, providing a unique addition to the existing literature.

Our results are limited by the use of a single, subjective health measure. Due to the need for brevity of the surveys, more comprehensive, objective or specific questions were not included, or they were not comparable between Ukrainian and non-Ukrainian samples. While self-rated health is highly predictive of morbidity and mortality, its validity in cross-national comparisons has been challenged because definitions of what "health" entails are culturally specific [33]. Furthermore, interpreting the results of this measure in the context of migration research is complicated by the fact that frames of reference of what constitutes good health may change during and after the migration process. We believe that these issues play a minor role the analysis of the Ukrainian sample, as respondents all migrated from the same country of origin at the same point in time. However, the high variability observed in the analysis of refugees from other countries of origin may reflect the subjective and context-dependent nature of the outcome measure. Our analysis is further limited by treating all non-Ukrainian refugees as a single group, although this group may be highly heterogeneous. While the majority of the non-Ukrainian sample arrived from Syria during the civil war, other countries of origin have also been included in the analysis and cannot be investigated separately due to low sample sizes. Nevertheless, the sample broadly captures individuals arriving under similar legal and social conditions, which was the focus of the present analysis. A further limitation is the cross-sectional nature of our study design, which does not allow for causal inference and entails the risk of reverse causality. Further research should employ longitudinal study designs, including the collection of health information prior to flight, to fully capture changes in health status throughout the migration trajectory.

Our results have several implications for the policy response to refugee migration in Europe. While the EU Temporary Protection Directive has been instrumental in ensuring ease of participation in German society for Ukrainian refugees, on its own it is not sufficient to ensure the health care needs of Ukrainians are met. Our research demonstrates that some Ukrainians continue to face difficulties in their life in Germany, and that such difficulties are strongly correlated with subjective health. In addition to affording access to legal, social, and economic systems, access to the health system must be ensured by adopting established models of care for refugee and migrant populations, including participative health communication strategies, the expansion of medical interpretation services and outreach services in the community. For non-Ukrainian refugees, our findings reinforce that restrictive legal conditions delay—but do not eliminate—the health impact of post-migration factors. These barriers to social and economic participation pose substantial risks for the health of affected individuals and the health system as a whole [18,19], although such risks may only be evident at a later stage of the integration process. Swift access to needs-based and diversity-competent health care [34] should therefore be a priority for all refugees, not just those from Ukraine.

## Supporting information

**S1 File.  Supplementary information.**
(DOCX)

## Author contributions

**Conceptualization:** Louise Biddle.

**Data curation:** Andrea Marchitto.

**Formal analysis:** Andrea Marchitto.

**Funding acquisition:** Sabine Zinn.

**Investigation:** Louise Biddle, Andrea Marchitto.

**Methodology:** Louise Biddle, Andrea Marchitto, Sabine Zinn.

**Supervision:** Sabine Zinn.

**Visualization:** Louise Biddle, Andrea Marchitto.

**Writing – original draft:** Louise Biddle.

**Writing – review & editing:** Sabine Zinn.

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
