## [Decision Letter · Decision Letter 0]

16 Feb 2025

PGPH-D-24-02903

Pre- and post-migration determinants of self-rated health among Ukrainian refugees in Germany: a cross-sectional comparative analysis with recently arrived refugees from other countries of origin

Dear Dr. Biddle,

Thank you for submitting your manuscript to PLOS Global Public Health. After careful consideration, we feel that it has merit but does not fully meet PLOS Global Public Health’s publication criteria as it currently stands. Therefore, we invite you to submit a revised version of the manuscript that addresses the points raised during the review process.

We look forward to receiving your revised manuscript.

Kind regards,

Raquel Muñiz-Salazar, Ph.D.

Academic Editor

Journal Requirements:

2. Figures 1 and 2: please (a) provide a direct link to the base layer of the map (i.e., the country or region border shape) and ensure this is also included in the figure legend; and (b) provide a link to the terms of use / license information for the base layer image or shapefile. We cannot publish proprietary or copyrighted maps (e.g. Google Maps, Mapquest) and the terms of use for your map base layer must be compatible with our CC-BY 4.0 license. 

Additional Editor Comments (if provided):

The study presents an important and timely contribution to the field of refugee health by examining the determinants of self-rated health among Ukrainian and non-Ukrainian refugees in Germany. The use of a unique dataset and the focus on pre- and post-migration factors offer valuable insights with potential policy implications. The paper is well-structured, clearly written, and relevant to a broad audience interested in migration, public health, and integration processes.

However, the reviewers recomment the following to strengthen the manuscript:

Deepen the Comparative Analysis

Contextualize with Past Migration Trends

Enhance Methodological Transparency

Clarify the rationale behind the threshold used for the Political Terror Scale (PTS).

Explore the "integration paradox" in greater depth, considering why initial integration efforts might not directly translate into better health outcomes.

Provide more concrete policy recommendations related to healthcare access, mental health support, and language-focused interventions.

Reviewers' comments:

Reviewer's Responses to Questions

**Comments to the Author**

1. Does this manuscript meet PLOS Global Public Health’s publication criteria? Is the manuscript technically sound, and do the data support the conclusions? The manuscript must describe methodologically and ethically rigorous research with conclusions that are appropriately drawn based on the data presented.

Reviewer #1: Yes

Reviewer #2: Yes

2. Has the statistical analysis been performed appropriately and rigorously?

Reviewer #1: Yes

Reviewer #2: No

3. Have the authors made all data underlying the findings in their manuscript fully available (please refer to the Data Availability Statement at the start of the manuscript PDF file)?

Reviewer #1: No

Reviewer #2: Yes

4. Is the manuscript presented in an intelligible fashion and written in standard English?

Reviewer #1: Yes

Reviewer #2: Yes

5. Review Comments to the Author

Reviewer #1: The theme of the article is topical in recent years and this kind of research is needed. Authors used unique datasets on refugees in Germany. I am convinced that the paper presented would be of interest to a wide range of readers. The paper is well written and well structured. However, I have a couple of comments.

1) Could the authors deeper collaborate on differences between Ukrainian and non-Ukrainian refugees and how the characteristics of these two groups could potentially affect the results? The Ukrainian refugees are mostly highly educated women compared to non-Ukrainian refugees mainly men with low education. These are two very different groups of refugees. Ukrainian refugees - women, mostly coming with small children, which may hinder their access to labor market + highly educated women, which may lead to overqualification on the labor market leading to worsening the self-rated health. On the other hand, learning German will be easier for highly educated people. In contrast, non-Ukrainian refugees – men without family obligations, but low educated with less capability to learn German and integrate in the society / labor market. Of course, these different characteristics may lead to different outcomes in terms of self-rated health and their pre- /post-migration determinants.

2) It would be interesting to have some information on the past migration trends, i.e. describe briefly the flow of economic migrants from the given countries (Ukraine, Syria, Iraq, Afghanistan, etc.) before the war conflicts to have an idea about the diaspora network. It was mentioned that lack of contact, isolation, discrimination is a problem. What is the role of diaspora in refugees’ settlement and the consequences on their health outcomes?

3) Although the main idea was to compare the Ukrainian refugees and non-Ukrainian refugees in Germany, it would be interesting to have some comparison on the determinants of health among Ukrainian refugees in different countries. For instance, if the results for Germany hold also for Poland, Slovakia, Czechia, Austria etc. Or if the Ukrainian refugees differ by host country. There are some publications on a similar topic that could serve as a basis for brief discussion.

- Kohlenberger et al. High self-selection of Ukrainian refugees into Europe: evidence from Krakow and Vienna. Plos One. 2023;18(12):e0279783

- Labberton et al. Trends in the health status of Ukrainina refugees in Norway according to month of arrival during 2022. BMC Public Health. 2024;24(1):3127

- Mazhak et al. Self-reported health and coping strategies of Ukrainian female refugees in the Czech Republic. European Societies. 2024;26(2):411-437

- Kulhánová et al. Determinants of self-rated health among highly educated Ukrainian women refugees in Czechia: analysis based on cross-sectional study in 2022. BMC Women’s Health. 2024;24(1):206

4) Comments related to a formal style of the manuscript:

- I guess the authors swapped figure 1 and figure 2.

- Would it be possible to include the name (or at least the abbreviation) of the Ukrainian regions in the maps in figure 1 and 2? Not all readers have to be familiar with the regional geography of Ukraine. Being able to read the map from your article without going into other literature or internet sources would be helpful. In light of other studies, it would also be helpful for comparison of regions from other publications.

- In figures 3 and 4, it would be nice to have a clear division of variables belonging to sociodemographic, pre-migration and post-migration factors. I know you described that in the text; however, the figures should be self-explanatory. Drawing a line between the groups of factors would be helpful.

- Figure 4 should give results for non-Ukrainian refugees, then I do not understand the variable “reason for leaving Ukraine” among pre-migration factors in this regression. Could the authors please explain if this is correct? Or corrected if this was a mistake?

Reviewer #2: REVIEWER COMMENTS

This is an important and interesting study where the authors examined the determinants of self-rated health outcomes amongst Ukrainian refugees in Germany: pre-and-post migration. Authors highlighted that experience of discrimination, and social isolation were critical post-migration factors affecting refugees’ health but those who made attempts to integrate by learning the German language or becoming proficient in the German language have better post-migration health outcomes. These findings are important and highlight the areas in refugee health that might need policy interventions.

This work is well written and presented. However, I have few comments for the attention of the authors.

INTRODUCTION

The introduction provides a solid background, but it could be more concise. Reducing redundancy (e.g., repeated mentions of the EU Temporary Protection Directive) would enhance readability.

METHODOLOGY

The inclusion of the Political Terror Scale (PTS) and war intensity metrics should be better justified and why this threshold of (>3)? Why not use individual-level trauma data?

Statistical Analysis

I suggest the authors provide more details on missing data patterns and whether MI results differ from complete case analysis. The study uses multiple imputations (MI) to address missing data, which is a valid approach. However, it would be useful to provide more details on: A) missing data patterns-were certain variables missing more frequently? If so, were they missing at random (MAR) or not at random (MNAR)? ; B) Impact of MI-while the Fraction of Missing Information (FMI) was reported (3.5% for Ukrainians, 6.1% for non-Ukrainians), further sensitivity analysis comparing complete case analysis vs. imputed results could add credibility.

I suggest additional model specifications and fit measures. The reported Variance Inflation Factors (VIFs) suggest no severe collinearity (VIFs range from 1.01–1.72 for Ukrainians, 1.04–1.49 for non-Ukrainians), but it's unclear whether interaction terms were tested. The model fit is evaluated using an F-test (which is appropriate for linear regression but less common for logistic regression). Consider adding: Hosmer-Lemeshow test to check calibration and Pseudo R² (e.g., McFadden’s R²) to assess explanatory power.

I suggest authors should conduct interaction tests for gender, age, and social isolation to check for differential effects. The logistic regression results indicate that post-migration factors (e.g., discrimination, social isolation, language proficiency) are significantly associated with health outcomes in the Ukrainian sample but not in the non-Ukrainian sample (except for social isolation). It’s unclear whether interactions (e.g., between gender and social isolation) were tested. If not, it would be valuable to explore whether certain effects are gender-dependent or vary by age groups.

I suggest stratified analysis by education/employment level to address potential selection bias. The Ukrainian sample is highly educated (70.6% with high education), and 84.4% were employed before flight, which is significantly different from the non-Ukrainian refugee sample (where 60.6% had low education, and only 60.8% were employed pre-migration). Could this education/employment difference bias the health comparisons? A stratified analysis (by education level or employment status) could ensure robustness.

I suggest further sensitivity analysis. The manuscript includes several sensitivity checks, which is a strength. However, the alternative health measure (worries about health) led to some changes in results, suggesting the primary self-rated health measure might not fully capture health conditions. The Political Terror Scale (PTS) vs. war fatalities measure did not show significant associations, but it would be useful to check whether those who fled later in the war (after prolonged exposure) had different health outcomes.

DISCUSSION

Some arguments could be more explicitly tied to the results. Consider structuring the discussion to first summarize findings, then compare them to existing literature, and finally discuss policy implications. For example, discussing (1) key findings, (2) comparison with prior research, (3) policy relevance.

It will be valuable to discuss more explicitly the EU Temporary Protection Directive’s limitations and suggest practical steps (e.g., targeted mental health services, language-sensitive healthcare, better social integration strategies).

Consider addressing the “integration paradox” in more depth. Why does early integration not necessarily translate to better health outcomes? How can policies mitigate this?

POTENTIAL LIMITATIONS

Self-rated health bias: Acknowledge potential cultural differences in self-assessing health and explore how this might affect comparisons between Ukrainian and non-Ukrainian refugees.

Cross-sectional study design: Acknowledge that this study does not establish causality and suggest longitudinal follow-ups to track evolving health conditions.

CONCLUSION

The conclusion touches on policy relevance but could be more specific. Suggest concrete measures for improving healthcare access, addressing discrimination, and integrating mental health support.

6. PLOS authors have the option to publish the peer review history of their article (what does this mean?). If published, this will include your full peer review and any attached files.

**Do you want your identity to be public for this peer review?** For information about this choice, including consent withdrawal, please see our Privacy Policy.

Reviewer #1: No

Reviewer #2: **Yes: **Dr. Barnabas Bessing

---

## [Decision Letter · Decision Letter 1]

29 Jul 2025

PGPH-D-24-02903R1

Pre- and post-migration determinants of self-rated health among Ukrainian refugees in Germany: a cross-sectional comparative analysis with recently arrived refugees from other countries of origin

Dear Dr. Zinn,

Thank you for submitting your manuscript to PLOS Global Public Health. After careful consideration, we feel that it has merit but does not fully meet PLOS Global Public Health’s publication criteria as it currently stands. Therefore, we invite you to submit a revised version of the manuscript that addresses the points raised during the review process.

We look forward to receiving your revised manuscript.

Kind regards,

Raquel Muñiz-Salazar, Ph.D.

Academic Editor

Journal Requirements:

Additional Editor Comments:

While the manuscript demonstrates methodological rigor, particularly in its statistical approach and discussion of results, several key concerns warrant attention before publication.

The short title fails to reflect the comparative nature of the study, which may mislead readers regarding its scope. Furthermore, the rationale for comparing Ukrainian refugees with those from other countries is insufficiently justified, both in the abstract and throughout the paper.

This raises concerns about the relevance and interpretability of the analysis, given the substantial demographic and contextual differences between the two groups.

Minor inconsistencies in data reporting should also be corrected.

Overall, revision is recommended to clarify the study’s comparative intent, justify its analytical framework, and ensure internal consistency.

Reviewers' comments:

Reviewer's Responses to Questions

**Comments to the Author**

1. If the authors have adequately addressed your comments raised in a previous round of review and you feel that this manuscript is now acceptable for publication, you may indicate that here to bypass the “Comments to the Author” section, enter your conflict of interest statement in the “Confidential to Editor” section, and submit your "Accept" recommendation.

Reviewer #2: All comments have been addressed

Reviewer #3: (No Response)

2. Does this manuscript meet PLOS Global Public Health’s publication criteria? Is the manuscript technically sound, and do the data support the conclusions? The manuscript must describe methodologically and ethically rigorous research with conclusions that are appropriately drawn based on the data presented.

Reviewer #2: Yes

Reviewer #3: Partly

3. Has the statistical analysis been performed appropriately and rigorously?

Reviewer #2: Yes

Reviewer #3: Yes

4. Have the authors made all data underlying the findings in their manuscript fully available (please refer to the Data Availability Statement at the start of the manuscript PDF file)?

Reviewer #2: Yes

Reviewer #3: Yes

5. Is the manuscript presented in an intelligible fashion and written in standard English?

Reviewer #2: Yes

Reviewer #3: Yes

6. Review Comments to the Author

Reviewer #2: This is an important and timely manuscript that adds significant value to the literature on refugee health, particularly in the context of the Ukraine crisis. The use of nationally representative datasets, comparative design, and robust statistical methods strengthens the credibility of the findings. The manuscript is well-written, theoretically grounded, and policy-relevant.

Reviewer #3: The shortened title, “Determinants of health among Ukrainian refugees in Germany,” is clear and concise but does not capture the comparison between the two refugee groups. The short title omits any mention of this comparative aspect, which should be reflected to avoid potential ambiguity about the article’s content.

Abstract. The contrast between the two groups is presented in a mechanical rather than a critical manner, and the rationale for comparing refugees from Ukraine with those from other countries is not clearly justified, raising questions about its relevance for publication in a prestigious journal.

Introduction. I recommend adding a citation to the study by Kulhanova et al. (2022) entitled "Determinants of self-rated health among highly educated Ukrainian refugee women in the Czech Republic: an analysis based on a cross-sectional study in 2022". This study provides an important and timely perspective on the determinants of health within a specific subgroup of Ukrainian refugees - highly educated women - in a Central European context comparable to Germany (34 and 35, p. 4).

The statistical approach is well thought out, especially due to thorough sensitivity analyses and appropriate handling of missing data.

Results:

p13, line 258, 259: Most Ukrainian refugees have a high level of education (70.6%), but in Table 1 is 73.3%

p19, line 264, 265: The proportions by country of origin don't quite match the Table? It should refer to the "country of birth" section

p19, line 265: 57.8% should be 57.9% according to the Table

p19, line 268: 75,7 % or 62,2 %?

The discussion of results is well-founded and well-structured.

In conclusion, it remains unclear why the comparison between refugees from Ukraine and those from other countries is relevant, given the substantial differences in their legal statuses, migration contexts, and demographic characteristics. The Ukrainian refugee group is relatively homogeneous in terms of nationality, cultural background, timing of arrival, and migration motivation, whereas the non-Ukrainian refugee group is heterogeneous, comprising individuals from diverse countries of origin, cultural backgrounds, lengths of stay, and reasons for migration. These demographic disparities complicate direct comparison and raise questions about the appropriateness and interpretability of the comparative analysis presented.

7. PLOS authors have the option to publish the peer review history of their article (what does this mean?). If published, this will include your full peer review and any attached files.

**Do you want your identity to be public for this peer review?** For information about this choice, including consent withdrawal, please see our Privacy Policy.

Reviewer #2: **Yes: **Dr.Barnabas Bessing

Reviewer #3: No

---

## [Editor Report · Decision Letter 2]

1 Sep 2025

PGPH-D-24-02903R2

Pre- and post-migration determinants of self-rated health among Ukrainian refugees in Germany: a cross-sectional comparative analysis with recently arrived refugees from other countries of origin

Dear Dr. Zinn,

Thank you for submitting your manuscript to PLOS Global Public Health. After careful consideration, we feel that it has merit but does not fully meet PLOS Global Public Health’s publication criteria as it currently stands. Therefore, we invite you to submit a revised version of the manuscript that addresses the points raised during the review process.

We look forward to receiving your revised manuscript.

Kind regards,

Raquel Muñiz-Salazar, Ph.D.

Academic Editor

Journal Requirements:

Additional Editor Comments (if provided):

The current short title is clear but omits the comparative aspect of the study. We recommend revising it to better reflect the analysis of both Ukrainian and non-Ukrainian refugee groups.

A central concern is the lack of justification for comparing these two groups, given their significant differences in legal status, migration context, and demographics. Clarifying the rationale for this comparison—both in the abstract and introduction—is essential to support the relevance and interpretability of your findings.

The reviewer also suggests citing Kulhanova et al. (2022), which provides useful context on Ukrainian refugees in a similar European setting.

Your statistical approach was well-received, particularly the handling of missing data and sensitivity analyses. However, please ensure consistency between the results reported in the text and the data in Table 1.

Lastly, we encourage you to address the implications of comparing two demographically distinct groups more explicitly in the discussion and conclusion.
---

## [Decision Letter · Decision Letter 3]

13 Oct 2025

Pre- and post-migration determinants of self-rated health among Ukrainian refugees in Germany: a cross-sectional comparative analysis with recently arrived refugees from other countries of origin

PGPH-D-24-02903R3

Dear Zinn,

We are pleased to inform you that your manuscript 'Pre- and post-migration determinants of self-rated health among Ukrainian refugees in Germany: a cross-sectional comparative analysis with recently arrived refugees from other countries of origin' has been provisionally accepted for publication in PLOS Global Public Health.

Best regards,

Raquel Muñiz-Salazar, Ph.D.

Academic Editor

All comments have been satisfactorily addressed; therefore, the final decision is to ACCEPT the manuscript for publication.

Reviewer Comments (if any, and for reference):

Reviewer's Responses to Questions

**Comments to the Author**

1. If the authors have adequately addressed your comments raised in a previous round of review and you feel that this manuscript is now acceptable for publication, you may indicate that here to bypass the “Comments to the Author” section, enter your conflict of interest statement in the “Confidential to Editor” section, and submit your "Accept" recommendation.

Reviewer #4: All comments have been addressed

2. Does this manuscript meet PLOS Global Public Health’s publication criteria? Is the manuscript technically sound, and do the data support the conclusions? The manuscript must describe methodologically and ethically rigorous research with conclusions that are appropriately drawn based on the data presented.

Reviewer #4: Yes

3. Has the statistical analysis been performed appropriately and rigorously?

Reviewer #4: No

4. Have the authors made all data underlying the findings in their manuscript fully available (please refer to the Data Availability Statement at the start of the manuscript PDF file)?

Reviewer #4: Yes

5. Is the manuscript presented in an intelligible fashion and written in standard English?

Reviewer #4: Yes

6. Review Comments to the Author

Reviewer #4: I think the article makes an interesting contribution to the literature and is written up appropriately, including strenghts and limitations of the data and methods.

7. PLOS authors have the option to publish the peer review history of their article (what does this mean?). If published, this will include your full peer review and any attached files.

**Do you want your identity to be public for this peer review?** For information about this choice, including consent withdrawal, please see our Privacy Policy.

Reviewer #4: No
